# CAT2VEC: LEARNING DISTRIBUTED REPRESENTATION OF MULTI-FIELD CATEGORICAL DATA

**Ying Wen, Jun Wang**
University College London, UK
MediaGamma Ltd, UK
{ying.wen,jun.wang}@cs.ucl.ac.uk

**Tianyao Chen, Weinan Zhang**
Shanghai Jiao Tong University
Shanghai, China
{tychen,wnzhang}@apex.sjtu.edu.cn

## ABSTRACT

This paper presents a method of learning distributed representation for multi-field categorical data, which is a common data format with various applications such as recommender systems, social link prediction, and computational advertising. The success of non-linear models, e.g., factorisation machines, boosted trees, has proved the potential of exploring the interactions among inter-field categories. Inspired by Word2Vec, the distributed representation for natural language, we propose Cat2Vec (categories to vectors) model. In Cat2Vec, a low-dimensional continuous vector is automatically learned for each category in each field. The interactions among inter-field categories are further explored by different neural gates and the most informative ones are selected by pooling layers. In our experiments, with the exploration of the interactions between pairwise categories over layers, the model attains great improvement over state-of-the-art models in a supervised learning task, e.g., click prediction, while capturing the most significant interactions from the data.

## 1 INTRODUCTION

There are different abstraction levels within data. For the low-abstraction continuous sensory data (such as images, videos, and audio) directly acquired from the physical world, quite often, the strong correlations (local patterns) are known a priori within the data. As such, one can directly embed the prior knowledge into a learning model such as neural networks to automatically distil such patterns and perform predictions (Krizhevsky et al., 2012; Graves et al., 2013). However, on the other hand, for high-abstraction data from our social and business activities, such as natural language and transnational log data, the data is commonly discrete and contains atomic symbols, whose meaning and correlation are unknown a priori. A typical solution is to employ embedding techniques (Bengio et al., 2003; Mikolov et al., 2013) to map the discrete tokens into a (low-dimensional) continuous space and further build neural networks to learn the latent patterns.

Multi-field categorical data is a type of high-abstraction data where the categories in each field are heterogeneous with those in other fields. Such a type of data is very widely used in data mining tasks based on transaction logs from many social or commercial applications, such as recommender systems, social link prediction, and computational advertising. Table 1 gives an example of multi-field categorical data in user behaviour targeting where we observe user browsing patterns, and given those multi-field categorical features, a common task is to predict their actions such as clicks and conversions (Zhang et al., 2014; Liao et al., 2014; Yuan et al., 2013).

As there is no explicit dependency among these inter-field categories, two solutions are mainly used for building machine learning models that extract the local patterns of the data and make good predictions. The first solution is to create combining features across fields, such as CITY:SHANGHAI&WEEKDAY:FRIDAY (Chapelle et al., 2015). Such feature engineering is expensive on human efforts and feature/parameter space. The second solution is to build functions (Rendle, 2012) or neural networks based on the feature embeddings (Zhang et al., 2016). These solutions are of low efficiency because of the brute-force feature engineering or aimless embedding interactions.

Table 1: A simple example of multi-field categorical data from iPinYou dataset (Liao et al., 2014).

| TARGET | GENDER | WEEKDAY | CITY | BROWSER |
|---|---|---|---|---|
| 1 | MALE | TUESDAY | BEIJING | CHROME |
| 0 | FEMALE | MONDAY | SHANGHAI | IE |
| 1 | FEMALE | TUESDAY | HONGKONG | IE |
| 0 | MALE | TUESDAY | BEIJING | CHROME |
| NUMBER OF CATEGORY | 2 | 7 | 351 | 6 |

In this paper, we propose an unsupervised pairwise interaction model to learning the distributed representation of multi-field categorical data. The interactions among inter-field categories are explored by different neural gates and the informative ones are selected by $K$-max pooling layers. Note that the $K$-max pooling process acts like the classic Apriori algorithm in frequent itemset mining and association rule learning (Agrawal et al., 1994). Repeating this pairwise interaction with $K$-max pooling, our Cat2Vec model automatically extracts salient feature interactions and further explores higher-order interactions.

To train the pairwise interaction Cat2Vec model effectively, we present a discriminant training method to estimate the category vectors. Furthermore, with the exploration of the pairwise and high-order category interactions, our Cat2Vec model attains great performance improvement over state-of-the-art models in supervised learning tasks, such as user response rate prediction, while successfully captures the most significant interactions in unsupervised learning tasks.

## 2    RELATED WORK AND PRELIMINARIES

In this section, we outline the major data representation methods that are used for representing the discrete categorical data. These methods serves as the preliminaries of our Cat2Vec model.

### 2.1    ONE-HOT REPRESENTATION

It is common to use one-hot representation for discrete data in natural language processing or computational advertising tasks. For the first data sample as an example, the data is vectorised by one-hot encoding as

$$\underbrace{[0,1]}_{\text{GENDER:MALE}}, \underbrace{[0,1,0,0,0,0,0]}_{\text{WEEKDAY:TUESDAY}}, \underbrace{[0,\dots,0,1,0,\dots,0]_{351}}_{\text{CITY:BEIJING}}, \underbrace{[1,0,0,0,0,0]}_{\text{BROWSER:CHROME}}. \tag{1}$$

With each category as a dimension, one-hot representation preserves full information of the original data. Two main problems of one-hot representation are that (i) it may suffer from the curse of dimensionality, especially in deep learning-related applications; (ii) it cannot capture the similarity of each word/category pair, and we cannot even find any relationships among the synonyms or categories in the same field.

### 2.2    DISTRIBUTED REPRESENTATION

Distributed representation is first proposed by Hinton (1986). The basic idea of distributed representation is training the model to map each word into a $d$-dimension vector (generally, $d$ is the hyperparameter of the model, and $d$ is far smaller than whole vocabulary size $N$ of words/categories), and the semantic similarity between the words/categories can be measured through the distance (such as cosine similarity, Euclidean distance) of their corresponding low dimension vectors. The Word2Vec (Mikolov et al., 2013) is one of the most common methods to train the distributed word vector representation. Compared with text, with the local patterns among the neighbour words, multi-field categorical data has no explicit order relationships among inter-field categories. Also, the text vocabulary size ($10^5$) is often much smaller than the category size ($10^6 \sim 10^8$), making our problem more difficult. Another difference between our Cat2Vec and Word2Vec is that Cat2Vec does not take the order into account or use any sliding window for context; in other words, we take all categories in the same training sample as the neighbour of a category.

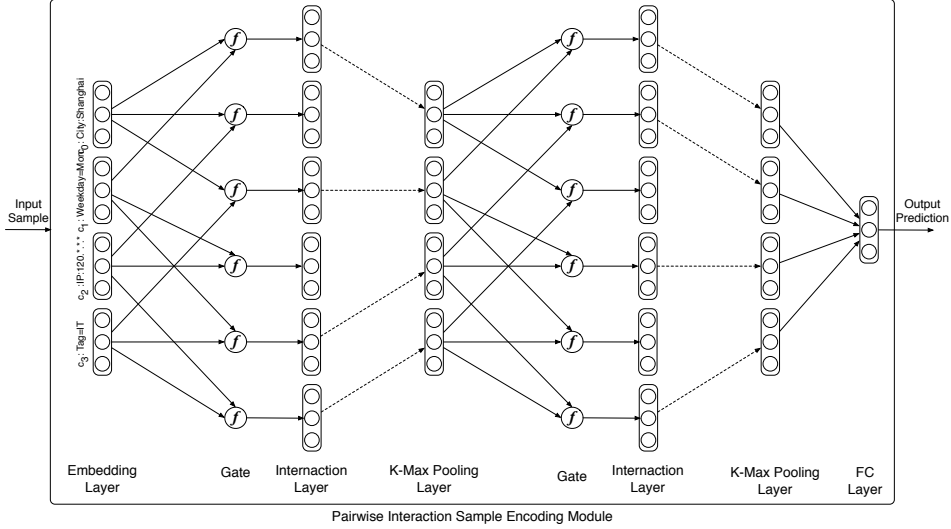

Figure 1: The proposed sample encoding module. At first, each category pair will be fed into a gate to get the interaction between two categories. Next, using K-max pooling to capture important interactions. Repeat above two steps, which could capture higher level category interactions. Finally, we use a full connection layer to transform final interaction vectors into the prediction.

## 3    PAIRWISE INTERACTION CAT2VEC MODEL

In this section, we introduce a pairwise interaction Cat2Vec model and its training method in detail. We design neural gates in the model to capture the interactions between each pair of categories, followed by the $K$-max pooling layers to select the most important interactions. We then repeat this processes to explore higher level interactions. Figure 1 illustrates the overview of the proposed architecture.

### 3.1    INTERACTION AND POOLING LAYERS

**Interaction Layer.** To evaluate the interaction between each pair of categories, we use a gate to obtain the interaction result. Mathematically, a gate is a function $f : \mathbb{R}^d \times \mathbb{R}^d \to \mathbb{R}^d$ that takes any pair of category vectors $c_i$ and $c_j$ in the same sample $c$ as input, and outputs interaction result vector $c'_{i,j} = f(c_i, c_j)$. The interaction output vector $c'_{i,j}$ acts as a certain combining feature of $c_i$ and $c_j$. Note that $c'_{i,j}$ keeps the same dimension as the category embedding vectors like $c_i$ and $c_j$ so that it can be further used to interact with other categories.

We provide several options of gate $f$ as:

$$f^{\text{sum}}(c_i, c_j) = c_i + c_j, \tag{2}$$

$$f^{\text{mul}}(c_i, c_j) = c_i \odot c_j, \tag{3}$$

where $\odot$ is the element-wise multiplication operator. We can also can employ more complex gates, such as the highway gate (Srivastava et al., 2015), which is formulated as

$$f^{\text{highway}}(c_i, c_j) = \tau \odot g(\mathbf{W}_H(c_i + c_j) + b_H) + (1 - \tau) \odot (c_i + c_j), \tag{4}$$

where $g$ is a nonlinear function and $\tau = \sigma(\mathbf{W}_\tau(c_i + c_j) + b_\tau)$ represents a "transform gate".

Then we apply the gate $f$ on each pair of category vectors $c_i, c_j$:

$$c' = [c'_{1,2}, c'_{1,3}, \cdots, c'_{1,n}, \cdots, c'_{n-2,n-1}, c'_{n-1,n}]. \tag{5}$$

After the interaction, an activation function will be applied to implement the non-liner transformation.

$K$**-Max Pooling Layer.** We next describe a pooling operation that is a generalisation of the max pooling based on the norm length of interaction outputs of each pair of category vectors. We keep

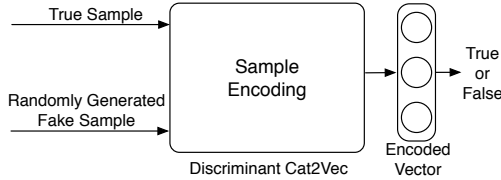

Figure 2: The discriminant Cat2Vec model which learns the category embedding by training a discriminator to distinguish the true samples from the fake ones.

the $K$ maximum interaction output vectors $c'_{i,j}$ according to their norm length, where $K$ is the number of the original categories of the training sample. It would keep the max-pooling result $c'_{\text{kmax}} = [c'_1, c'_2, \cdots, c'_K]$ having the same size with the original embedding matrix $c$ and $c'_K$ is the embedding vector in $c'$ in Eq. (5) that has top-$K$ normal length.

Before producing an output for the interaction results, the interaction and $K$-max pooling operations will be repeated for several times in order to capture high-level interactions among the different field category vectors. After that, we output a prediction from the final interaction vector representation by a fully connected layer. Note that the above network structure can be used to build an auto-encoder to conduct unsupervised learning (Vincent et al., 2008). We leave this for future work, while staying with the label output network for both supervised (containing both negative and positive examples) and unsupervised (only containing positive examples where negative examples are generated randomly) learning tasks.

An interesting discussion is to compare our Cat2Vec model with association rules mining, which aims to identify the most frequently appeared joint category instances (items), with or without a condition. *Apriori* (Agrawal et al., 1994) is a popular algorithm for association rules mining by exploiting dependencies between candidate frequent itemsets of length $K$ and frequent itemsets of length $K - 1$. In our pairwise interaction Cat2Vec model, with neural networks, we provide an alternative way of generating such high-order interactions (thus itemsets) among category instances. Via the pooling operation, our model can also find the most frequent category set automatically, which will be demonstrated and tested from our experiments in the following Sections 4 and 5.

## 3.2 DISCRIMINANT CAT2VEC - TRAINING METHOD OF PAIRWISE INTERACTION CAT2VEC

To train the pairwise interaction Cat2Vec model, we design a training scheme called discriminant Cat2Vec, which would train the model in a supervised way for unsupervised learning of the data.

In the discriminant Cat2Vec, we feed the Sample Encoding Module showed in Figure 1 with a true or fake sample, the encoded sample vector will be followed by an MLP to predict the probability $p$ of a true sample. As such, the generation of a fake sample would influence the learned category vector. In this paper, we generate a fake sample following this way: first, randomly choose a sample from the training set; second, randomly choose several categories in this sample and replace them with randomly chosen categories that belong to the same field. For example, we get a user behaviour instance $x = [\text{WEEKDAY:WEDNESDAY}, \text{IP:1.1.*.*}, \text{GENDER:MALE}, \text{CITY:BEIJING}]$, and we randomly choose the category CITY:BEIJING and replace it with CITY:SHANGHAI, then we build a fake sample $x' = [\text{WEEKDAY:WEDNESDAY}, \text{IP:1.1.*.*}, \text{GENDER:MALE}, \text{CITY:SHANGHAI}]$. The discriminant network is then trained to predict whether the new sample should be a true sample. The loss function of discriminant network is average cross entropy, which would maximise the likelihood of correct prediction:

$$ L = \frac{1}{M} \sum_{i=1}^{M} -y_i \log(p_i) - (1 - y_i) \log(1 - p_i), \tag{6} $$

where $M$ is the number of training samples. The $i$-th sample is labelled with $y_i \in \{1, 0\}$, which means true or fake sample, and $p_i$ is the predicted probability that the given training sample is true.

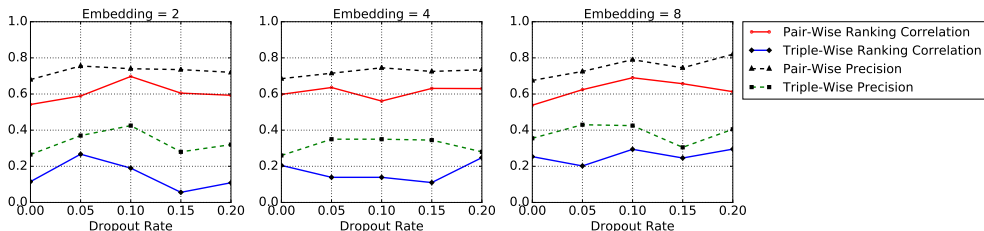

Figure 3: Precision and rank correlation on synthetic data, bigger embedding size and appropriate dropout rate leads to achieve better performance.

# 4 SYNTHETIC DATA EXPERIMENTS

To explore and add our understanding of the pairwise interaction Cat2Vec model, we conduct a simulation test with synthetic data. In particular, we are interested in understanding how the learned vectors would be able to capture and leverage the most significant patterns embedded in the data.

## 4.1 SYNTHETIC DATASET AND EVALUATION METRICS

To simulate the real-world multi-field categorical data, we use multivariate normal sampling to generate the true data distribution for the following experiments [1] . Suppose the data has 4 fields $\{A, B, C, D\}$, each field contains 10 categories, and a sample can be represented as $x = (a_i, b_i, c_i, d_i)$. We then randomly generate the means and covariance matrix for 4-dimensional truncated multivariate normal sampling with two-sided truncation. This sampling method can generate 4 float numbers between 0 and 10. We can convert the float numbers to integer which can represent the categories in 4 fields. In such a way, we can generate the data with specific joint distribution, which means certain categorical pair or 3-tuple like $p(a_4, b_4)$ or $p(a_3, c_5, d_6)$ may have a higher joint distribution probability. Recall that in our pairwise interaction Cat2Vec model, we have a $K$-max pooling layer, which will *select* the most popular category pairs in the dataset. Repeating the pairwise interaction layers and $K$-max pooling layers, we can also explore a high order categorical 3-tuple or 4-tuple etc. Therefore, our task here is to evaluate if our model would be able to capture these frequently occurred patterns from a given dataset; in other words, to test if our model would be able to keep the category pairs with the highest joint distribution probabilities in the $K$-max pooling results. This processes is in line with association rule mining (Agrawal et al., 1994), exploring the frequent categorical $n$-tuple from frequent categorical $(n-1)$-tuple.

We generate the positive data according to the above truncated multivariate normal sampling and choose uniform sampling to generate the fake (negative) data. We then apply discriminant Cat2Vec to train the model. Because we know the true distribution of the generated real data, the most frequent category pairs/triples are known. We use precision and Spearman's rank correlation coefficient to evaluate the results of 1st/2nd $K$-max pooling layer (category pairs/triples pooling results), to see if the model can learn the true joint distribution in the real data. The details of the evaluation metrics are described in the following section.

To evaluate how our network structure and $K$-max pooling help identify the significant $n$-tuples, we feed 1000 samples to the trained model and record the 1st and 2nd $K$-max pooling layers' results. Then we count the frequency of the category pairs/3-tuples in the real samples, and select top 20 ranked category pairs/3-tuples as target. Then we count the frequency of max-pooled category pairs/triples in the results and compare the top 20 frequent category pairs/3-tuples in the results to calculate precision and Spearman's rank correlation coefficient. Precision measures the fraction of category pairs/triples in the results that are also in the target. The Spearman's rank correlation coefficient measures the correlation between two ranked lists.

---

[1] the experiment code is available: `https://github.com/wenying45/cat2vec`

## 4.2 RESULT AND DISCUSSION

Figure 3 summarises the results of the precision and the rank correlation on synthetic data. We can see that our model can easily find over 80% of the category pairs with high joint distribution probabilities under the best parameter settings. From the rank correlation, our model can make the ranking correlation over 0.6 of category pairs which means the category pairs with higher joint distribution probability would be more possible to appear in the $K$-max pooling result. As for the category triples case, the precision and rank correlation become lower than the category pairs', because finding 3-order combination is harder and relies on the accuracy from the 2-order. We also vary the dropout rate against those measures. It shows that dropout tends to help improving the accuracy of captured patterns. This can be explained by considering the fact that dropout brings randomness into the selection and allows exploration. But the best dropout rate seems rather arbitrary and highly dependent on the other parameter settings.

## 5 REAL-WORLD DATA EXPERIMENTS

In this section, we continue our experiment using a real-world advertising dataset for click-through rate estimation[2]. The iPinYou dataset (Liao et al., 2014) is a public real-world display ad dataset with each ad display information and corresponding user click feedback (Zhang et al., 2014). This dataset contains around 19.5M ad display instances with 14.8k positive user feedback (click). Each instance has 23 fields, and we choose 18 fields of them which have categories with occurrence larger than 10.[3]

## 5.1 UNSUPERVISED LEARNING EXPERIMENT

We continue our study on the model's ability of capturing the most significant patterns as we described in Section 3.2. Because the iPinYou dataset contains the unencrypted fields and categories, e.g. city, region and tag, so we choose the iPinYou dataset which has been introduced above as real positive) data. As for the fake (negative) data, we randomly choose a sample in the iPinYou dataset and randomly replace some categories with other categories in the same field to generate the fake data, similar to what we have introduced in Section 3.2. We also set up two baseline models to compare the model accuracy performance: (i) DNN Concat model, which concatenates category embedding vectors to make prediction, and (ii) DNN Sum model, which sums up the category embedding vectors to make the prediction.

We have tried different parameter settings and the performance is measured by the accuracy of our model to predict real samples. We also calculate the rank correlation coefficient and the precision to evaluate our model the same as we described in Section 4.1.

### 5.1.1 RESULT AND DISCUSSION

From Table 2, we see that on the iPinYou dataset, our pairwise interaction models can achieve the accuracy of 85% which is about 1.7% improvement comparing with the simple DNN models. Even the worst case in our model is better than the DNN models' best case. It means our model can find the extra information during the interactions and the $K$-max pooling processes. In addition, the model with interaction times as 3 usually yields better performance than that with interaction times as 2, which may be due to the fact that the more interaction times capture higher-order interactions and help make more accurate predictions. But the model with different gate types does not lead to significant difference.

We next use the same evaluation metrics that described in Section 4.1 to test the ability of capturing data patterns. We find that in the real-world dataset, our model is still able to keep high precision and rank correlation and can achieve even better performance. The precision and rank correlation on category pairs are over 0.8 which is a 30% improvement comparing to the performance on synthetic

---

[2]http://data.computational-advertising.org/
[3]The selected fields are WEEKDAY, HOUR, USER AGENT, IP, REGION, CITY, AD EXCHANGE, DOMAIN, URL, AD SLOT ID, AD SLOT WIDTH, AD SLOT HEIGHT, AD SLOT VISIBILITY, AD SLOT FORMAT, AD SLOT FLOOR PRICE, CREATIVE ID, KEY PAGE URL, AND USER TAGS.

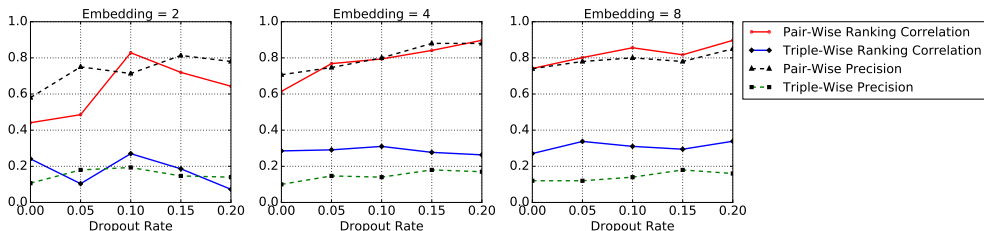

Figure 4: Precision and Rank Correlation on iPinYou Data; bigger embedding size and appropriate dropout rate leads to achieve better performance.

Table 2: Accuracy of distinguishing true impression from fake impression; embedding means embedding vector size and interaction is interaction times in our model.

| PARAMETERS | | GATE TYPE | | | DNN CONCAT | DNN SUM |
|---|---|---|---|---|---|---|
| | | SUM | MUL | HIGHWAY | | |
| embedding = 8 | interaction = 2 | 0.836 | 0.827 | 0.830 | 0.807 | 0.806 |
| | interaction = 3 | 0.831 | 0.834 | 0.830 | | |
| embedding = 16 | interaction = 2 | 0.838 | 0.836 | 0.837 | 0.828 | 0.817 |
| | interaction = 3 | 0.843 | 0.845 | 0.838 | | |
| embedding = 32 | interaction = 2 | 0.844 | 0.842 | 0.843 | 0.831 | 0.833 |
| | interaction = 3 | 0.848 | **0.850** | 0.843 | | |

dataset. For the category triples case, we also have similar performance compared with the synthetic dataset.

## 5.2 CLICK-THROUGH RATE PREDICTION EXPERIMENT

We now move to the evaluation on a supervised learning task. We consider click-through rate (CTR) prediction, which is important for many personalised Web services such as E-commerce, social recommendation and computational advertising (Yuan et al., 2013). The most widely used CTR estimation model is the logistic regression based on one-hot data representation. Many deep learning models have been further investigated in CTR prediction. Zhang et al. (2016) proposed Factorisation-Machine Supported Neural Networks (FNN) models for user response prediction. Convolutional Click Prediction Model (CCPM) (Liu et al., 2015) has been used in CTR prediction and gain some improvement on this task. To our knowledge, all of above previous work focuses on directly improving the prediction performance in supervised learning tasks and none of them investigates the learned representation of multi-field categorical data or how to learn the better representation.

In order to investigate our pairwise interaction model in the CTR task, we use the pairwise interaction sample encoding module to encode a training sample concatenated with the embedding vectors, which is followed by an MLP (multi-layer perceptron) to predict click-through probability. We choose following models as strong baselines:

- **Logistic Regression (LR)**: LR is a widely used linear model (Richardson et al., 2007).

- **Factorisation Machine (FM)**: Simply apply the factorisation machine on one-hot encoded sparse features of the training sample (Rendle, 2010).

- **CCPM**: CCPM (Liu et al., 2015) is a convolutional model for click prediction.

- **FNN**: A DNN model based on concatenated category vectors following with MLPs, being able to capture high-order latent patterns of multi-field categorical data (Zhang et al., 2016).

- **Cat2Vec-FNN-1**: This is our proposed architecture that only concatenates pairwise interaction output vectors among $K$-max pooling results to form the final vector representation and make prediction.

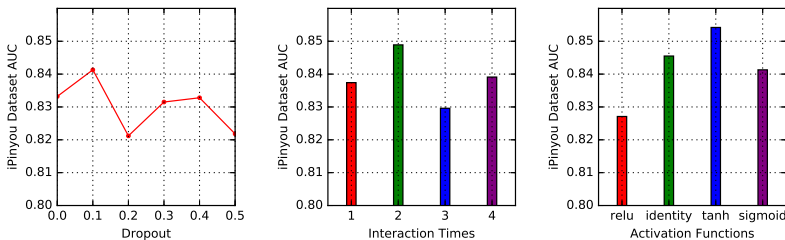

Figure 5: Performance Comparison over different Parameter Settings

Table 3: AUC of CTR prediction on iPinYou dataset.

| MODEL | LR | FM | CCPM | FNN | Cat2Vec-FNN-1 | Cat2Vec-FNN-2 |
|---|---|---|---|---|---|---|
| AUC | 0.8323 | 0.8349 | 0.8364 | 0.8453 | 0.8599 | **0.8640** |

- **Cat2Vec-FNN-2**: This is our proposed architecture that explore category vectors pairwise interaction result between $K$-max pooling results and category embeddings to form the final vector representation and make prediction.

We use Area Under ROC Curve (AUC) as the evaluation metrics to measure the performance of a prediction. Also we conduct the grid search for each model to make sure that each model has achieved its best performance. Specifically, empirically optimal hyperparameters are set as: the category embedding size is 16, the SGD batch size is 64, the Nadam (Sutskever et al., 2013) is set as SGD optimiser with default settings, the gate type is MUL and the norm type for $K$-Max Pooling is L2 norm, and the activation function as tanh. Then the model followed by three fully connected layer with width $[128, 32, 1]$. We also try different interaction times and finally set it as two (3-tuple), suggesting that a high order of interactions helps improve the performance, but more than two would overfit the data and thus managed the performance,

### 5.2.1 RESULT AND DISCUSSION

Table 3 gives the results of our CTR experiment, compared with various baselines. We see that there is about 3% improvement over LR. The AUC performance of the proposed Discrimination Cat2Vec models also outperforms the FM/CCPM/FNN model, as our model would be able to take higher order information into consideration, which helps make better decision.

In our pairwise interaction model, we also test different hyperparameters and settings, and the result is given in Figure 5. First, we evaluate the performance over different dropout rates, and find that setting dropout as 0.1 would be the best, as shown in Figure 5. We also explore the impact of interaction. From the result, the model with 2 interaction times would have better generalisation on the test set. Finally, we compare three different activation functions (sigmoid, tanh, relu) and set identity mapping as the baseline. The result shows that "tanh" yields the best performance, which has the advantages of non-linear transformation between $(-1, 1)$, and it may help gain more benefits on multi-field categorical data.

## 6 CONCLUSION

In this paper we have proposed a novel Cat2Vec model working on the multi-field categorical data. Different from the other models, Cat2Vec repetitively computes and selects inter-field category pairwise interactions to explore high-level interactions, which is analogous to the Apriori algorithm in association rule mining. Moreover, we present an efficient discriminant training method to estimate the category vectors. We also apply our pairwise interaction model on CTR prediction, of which we have observed a significant performance gain over several strong baselines. For future work, we plan to design more sophisticated gates to explore different interaction patterns among inter-field categories; also leveraging Cat2Vec in various data mining problems is of great interest to us.

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
