# Peer review of "Cat2Vec: Learning Distributed Representation of Multi-field Categorical Data"

_ICLR 2017 — rejected_

[Official Review · AnonReviewer3 · rating 4 · confidence 4 · 16 Dec 2016]
**A decent paper but with some issues.**

In this paper, the author proposed an approach for feature combination of two embeddings v1 and v2. This is done by first computing the pairwise combinations of the elements of v1 and v2 (with complicated nonlinearity), and then pick the K-Max as the output vector. For triple (or higher-order) combinations, two (or more) consecutive pairwise combinations are performed to yield the final representations. It seems that the approach is not directly related to categorical data, and can be applied to any embeddings (even if they are not one-hot). So is there any motivation that brings about this particular approach? What is the connection? 

There are many papers with similar ideas. CCPM (A convolutional click prediction model) that the authors have compared against, also proposes very similar network structure (conv + K-max + conv + K-max). In the paper, the author does not mention their conceptual similarity and difference versus CCPM. Compact Bilinear Pooling,

[Official Review · AnonReviewer2 · rating 5 · confidence 5 · 19 Dec 2016]
**An incremental paper with minor contributions and weak baselines**

The paper proposes a way to learn continuous features for input data which consists of multiple categorical data. The idea is to embed each category in a learnable low dimensional continuous space, explicitly compute the pair-wise interaction among different categories in a given input sample (which is achieved by either taking a component-wise dot product or component-wise addition), perform k-max pooling to select a subset of the most informative interactions, and repeat the process some number of times, until you get the final feature vector of the given input. This feature vector is then used as input to a classifier/regressor to accomplish the final task. The embeddings of the categories are learnt in the usual way. In the experiment section, the authors show on a synthetic dataset that their procedure is indeed able to select the relevant interactions in the data. On one real world dataset (iPinYou) the model seems to outperform a couple of simple baselines. 

My major concern with this paper is that their's nothing new in it. The idea of embedding the categorical data having mixed categories has already been handled in the past literature, where essentially one learns a separate lookup table for each class of categories: an input is represented by concatenation of the embeddings from these lookup table, and a non-linear function (a deep network) is plugged on top to get the features of the input. The only rather marginal contribution is the explicit modeling of the interactions among categories in equations 2/3/4/5. Other than that there's nothing else in the paper. 

Not only that, I feel that these interactions can (and should) automatically be learned by plugging in a deep convolutional network on top of the embeddings of the input. So I'm not sure how useful the contribution is. 

The experimental section is rather weak. They authors test their method on a single real world data set against a couple of rather weak baselines. I would have much preferred for them to evaluate against numerous models proposed in the literature which handle similar problems, including wsabie. 

While the authors argued in their response that wsabie was not suited for their problem, i strongly disagree with that claim. While the original wsabie paper showed experiments using images as inputs, their training methodology can easily be extended to other types of data sets, including categorical data. For instance, I conjecture that the model i proposed above (embed all the categorical inputs, concatenate the embeddings, plug a deep conv-net on top and train using some margin loss) will perform as well if not better than the hand coded interaction model proposed in this paper. Of course I could be wrong, but it would be far more convincing if their model was tested against such baselines.

[Official Review · AnonReviewer1 · rating 4 · confidence 4 · 28 Dec 2016]
**Weak comparison with baselines**

A method for click prediction is presented. Inputs are a categorical variables and output is the click-through-rate. The categorical input data is embedded into a feature vector using a discriminative scheme that tries to predict whether a sample is fake or not. The embedding vector is passed through a series of SUM/MULT gates and K-most important interactions are identified (K-max pooling). This process is repeated multiple times (i.e. multiple layers) and the final feature is passed into a fully connected layer to output the click prediction rate. 

Authors claim:
(1)	Use of gates and K-max pooling allow modeling of interactions that lead to state of art results. 
(2)	It is not straightforward to apply ideas in papers like word2vec to obtain feature embeddings and consequently they use the idea of discriminating between fake and true samples for feature learning. 

Theoretically convolutions can act as “sum” gates between pairs of input dimensions. Authors make these interactions explicit (i.e. imposed structure) by using gates. Now, the merit of the proposed method can be tested if a network using gates outperforms a network without gates. This baseline is critically missing – i.e. Embedding Vector followed by a series of convolution/pooling layers. 

Another related issue is that I am not sure if the number of parameters in the proposed model and the baseline models is similar or not. For instance – what is the total number of parameters in the CCPM model v/s the proposed model? 

Overall, there is no new idea in the paper. This by itself is not grounds for rejection if the paper outperforms established baselines.  However, such comparison is weak and I encourage authors to perform these comparisons.

[Final Decision · Program Chairs · 06 Feb 2017]
**ICLR committee final decision**

There is consensus among the three reviewers that (1) the originality of the proposed approach is limited and (2) the experimental evaluation is too limited in that it lacks strong baseline models as well as an ablation study that explores the different aspects of the proposed model.